# Diversity Learning Based on Multi-Latent Space for Medical Image Visual Question Generation

**DOI:** 10.3390/s23031057

**Published:** 2023-01-17

**Authors:** He Zhu, Ren Togo, Takahiro Ogawa, Miki Haseyama

**Affiliations:** 1Graduate School of Information Science and Technology, Hokkaido University, N-14, W-9, Kita-ku, Sapporo 060-0814, Hokkaido, Japan; 2Faculty of Information Science and Technology, Hokkaido University, N-14, W-9, Kita-ku, Sapporo 060-0814, Hokkaido, Japan

**Keywords:** visual question generation, medical image analysis, medical informatics, computer vision, natural language processing

## Abstract

Auxiliary clinical diagnosis has been researched to solve unevenly and insufficiently distributed clinical resources. However, auxiliary diagnosis is still dominated by human physicians, and how to make intelligent systems more involved in the diagnosis process is gradually becoming a concern. An interactive automated clinical diagnosis with a question-answering system and a question generation system can capture a patient’s conditions from multiple perspectives with less physician involvement by asking different questions to drive and guide the diagnosis. This clinical diagnosis process requires diverse information to evaluate a patient from different perspectives to obtain an accurate diagnosis. Recently proposed medical question generation systems have not considered diversity. Thus, we propose a diversity learning-based visual question generation model using a multi-latent space to generate informative question sets from medical images. The proposed method generates various questions by embedding visual and language information in different latent spaces, whose diversity is trained by our newly proposed loss. We have also added control over the categories of generated questions, making the generated questions directional. Furthermore, we use a new metric named similarity to accurately evaluate the proposed model’s performance. The experimental results on the Slake and VQA-RAD datasets demonstrate that the proposed method can generate questions with diverse information. Our model works with an answering model for interactive automated clinical diagnosis and generates datasets to replace the process of annotation that incurs huge labor costs.

## 1. Introduction

A question-and-answering (QA) system consists of a visual question-answering (VQA) model [1] and a visual question-generation (VQG) model [2]. VQA and VQG are interdisciplinary tasks incorporating computer vision [3] and natural language processing [4]. The VQA task is expected to answer an image-related question based on the content of the image, whereas the VQG task generates meaningful questions concerning the input image. QA systems have been extensively used in many fields, including retrieval systems [5,6] and medical image analysis [7,8]. In the field of medicine, a medical QA system, unlike other auxiliary clinical diagnosis systems [9], such as image segmentation [10], image classification [11], and surgical robots [12], can support the workload of performing radiographic image interpretation and pathological diagnosis simultaneously without dependence on expertise. Moreover, such a medical QA system would be helpful for patients to obtain reliable and accurate information even after treatment.

VQA models can assist in clinical diagnosis by answering questions given by physicians. However, the entire diagnostic process is still dominated by human physicians, and an interactive automated clinical diagnosis is required to address the current scarcity of medical resources. An interactive system consists of a question-and-answer format, which captures diverse information and adjusts the direction of the diagnosis by generating different questions. Compared with systems that individually answer questions given by physicians, automated access to questions can allow for less physician involvement and a more self-regulated diagnosis. Although the questions can be predefined, the number and variety of predefined questions will be limited to account for generalizability. Because patients’ conditions differ significantly, it is necessary to generate different questions from diverse perspectives for specific patients.

Generally, VQG is considered a more intelligent task than VQA because of the need for a deeper understanding of the information and for framing it in exact words, which prevents answers from being misconstrued [13]. Popular VQG frameworks consist of an encoder and a decoder. Various convolutional neural networks (CNNs), such as VGGNet [14] and ResNet [15], are generally used as encoders. On the other hand, long short-term memory (LSTM) [16] and bidirectional encoder representations from transformers (BERT) [17] are extensively used as decoders. Yang et al. [18] first used a recurrent neural network (RNN) [19] and a CNN in the VQG model. A template-based VQG method using a fixed object, attributes, and relations between a pair of objects was proposed by Geman et al. [20].

Medical VQG is a more challenging task than VQG in the general domain because questions generated from medical images require more information for an accurate diagnosis. To solve this problem, Sarrouti et al. [21] proposed a variational auto-encoder (VAE) [22] model to generate questions from radiology images. Many methods have been proposed in the “imageCLEF” [23,24], a medical image processing competition. Since 2020, medical VQG has been included in the competition, and many contestants have proposed methods for improving the accuracy of medical VQG.

Although VQG is a promising technology, there are still two major problems with state-of-the-art VQG methods. First, little attention is given to the informativeness of generated questions. Specifically, a single medical image contains a large amount of information, and to effectively utilize this information, questions with various meanings should be generated. However, during the question generation process for a single image, previous methods generated one question at a time and generated multiple questions through repeated sampling [18]. This approach frequently yielded generated questions with similar meanings. Although Krishna et al. [25] have proposed a model that maximizes the information in the image involved in the latent space, the question-generation process has not been improved. The second problem is the limitations of evaluation metrics. The standard evaluation metrics for VQG, such as BLEU [26] and METEOR [27], are primitively used in machine translation. These metrics cannot evaluate the semantic complexity and diversity of the results. Therefore, it is critical to introduce a more appropriate evaluation metric.

In this study, we propose a novel medical VQG model based on diversity learning with a multi-latent space. The proposed method also uses category information to control the category of generated questions. We simultaneously generate various question sets with low similarity instead of repeated sampling to improve the diversity of generated questions. Our diversity learning uses a multi-latent space to contain different aspects of information from the training data, and these latent spaces are used to generate meaningful question sets. Specifically, to train these latent spaces, we use the maximal information coefficient (MIC) loss to maximize their information difference. Furthermore, sentences with more information are thought to have lower semantic similarity than those with less information. Thus, to solve the second problem, we use a semantic model to calculate the average semantic similarity between generated questions and introduce similarity as a metric. Our model outperforms other models in terms of the diversity of generated questions and semantic similarity.

Finally, we summarize the contributions of this study.

We propose a novel medical VQG model based on diversity learning with a multi-latent space to solve the conventional informativeness problem of the VQG task and the multi-latent space is trained by the newly introduced MIC loss.We introduce semantic similarity as an evaluation metric of the medical VQG model to address the limitations of previous evaluation metrics.

The rest of the paper is organized as follows. Section 2 gives a brief description of the related work. Section 3 presents the proposed VQG model based on a multi-latent space. Section 4 presents the qualitative and quantitative experimental results. Section 5 discusses these results. Finally, we conclude with a summary in Section 6.

## 2. Related Works

The development of the VQA and VQG tasks has piqued the interest of researchers because of their broad application prospects in the medical field. Medical VQA and VQG systems aim to help clinical decision-making and provide patients with a better understanding of their illness to reduce patient anxiety and potential physician misunderstandings. Unlike other medical artificial intelligence applications, which are typically limited to predefined diseases or organ types, medical VQA and VQG can understand free-form questions in natural language and provide reliable and user-friendly answers.

Medical VQA and VQG tasks are technically considered more challenging than general domain VQA and VQG tasks because of the difficulty of constructing a large-scale medical dataset, which requires experts’ annotation, and the demand for disease-specific design.

### 2.1. Medical Visual Question Answering

At present, most of the medical VQA models are based on a general framework, the *joint embedding* [2,28], which consists of the following four modules: a visual encoder, a language encoder, a features joint module, and an answering module adapting to a specific task. Pre-trained CNNs, such as VGGNet [14] and ResNet [15], are frequently used as visual encoders that extract image features. Moreover, popular language coding models, such as LSTM [16], BERT [17], and Transformer [29], are extensively used as language encoders.

However, because architectures used in medical VQA research have fewer categories than those used in the general domain VQA, there remains a lot of research potential. Recently, architectures used in the general domain VQA, such as the compositional model neural module network [30], have not been introduced to the medical field but have significant potential to be explored in the future.

### 2.2. Medical Visual Question Generation

Traditional and novel generation models [31,32] are the two main architectures used for VQG models. Traditional models include rule-based models [33] and template-based models [20,34]. Rule-based models use the Stanford Parser [35] to generate simple sentences transformed from the description of the target image, which is used to find words that represent potential objects, relationships, locations, and colors. Furthermore, questions can be generated using the constituent “wh-” word and verbs to construct interrogative sentences. Real scenes typically have a high demand for the diversity of generated questions, whereas questions generated by traditional methods have similar structures and are very rigid. Novel generation models [21,25,36] are mostly composed of multiple modules, including an image encoder and a question decoder. The recent pre-trained visual backbone can extract convolutional features from images. Extracted images and answer features are then decoded using a language decoder to generate questions. We use the categories of questions in the medical domain to exert control over the generated results, an approach inspired by Uppal et al. [36] and Krishna et al. [25]. Furthermore, Krishna et al. [25] inspired us to construct latent spaces, which simplify the gradient optimization by replacing discrete vectors.

However, research on VQG in the medical field is still in the exploratory stage. Sarrouti et al. [21] proposed a novel generation model from radiology images. Although the VAE-based model proposed by Sarrouti et al. [21] lacks diversity, it demonstrates the feasibility of VQG in the medical field, which serves as a guide for our work. The methods [37,38,39] proposed in the “imageCLEF” are based on the novel generation model. These models focus on data processing rather than on their improvement, and there has been an insufficient investigation into a more effective model.

## 3. Model Framework

We provide an overview of our method in Figure 1. To more intuitively represent our model, we demonstrate the model with two latent spaces as an example. Inspired by the concept of VAE, which consists of an encoder and a decoder, we propose a network for learning the predicted distribution of the training data to approximate the true distribution. First, we use a CNN and a multilayer perceptron (MLP) as the encoder to obtain features from the input data, and question sets can be generated by the decoder LSTM. Our model comprises an image-category feature processor and a multi-latent space processor. The image-category feature processor introduces the category into the model, making the category of the generated question controllable. Furthermore, diversity learning is realized using the multi-latent space processor, which can obtain diverse information about different aspects of the training data. Question sets with diverse information can be generated using the different information aspects obtained in the latent spaces.

### 3.1. Image-Category Feature Processor

The image-category feature processor controls the category of the generated question by introducing the category feature into the model. For *n*-th training data (*n* = 1, 2, …, *N*; *N* being the number of training data), there is an input image $I_{n}$, a category word $C_{n}$, and a ground truth $G_{n}$. We define an image encoder CNN as $E_{I}\left(\cdot \right)$ and a category encoder MLP as $E_{C}\left(\cdot \right)$. The processor can output the image feature $f_{n}^{I}$ and the category feature $f_{n}^{C}$ as follows:(1)$$\begin{array}{cc} \; & f_{n}^{I}=\;E_{I}\left(I_{n}\right),\\ \; & f_{n}^{C}=\;E_{C}\left(T\left(C_{n}\right)\right), \end{array}$$
where T(·) represents an embedding layer to embed $C_{n}$ to a sentence vector. We concatenate $f_{n}^{I}$ and $f_{n}^{C}$ to obtain the joint vector as $f_{n}^{J}=\;\left[f_{n}^{I},f_{n}^{C}\right]$, and because the information for both the image and category is contained in $f_{n}^{J}$, the model can control the category of the generated question. The category information can be considered by concatenating the encoded features of the image and category to obtain the joint vector, which is the input of the multi-latent space processor.

### 3.2. Multi-Latent Space Processor

Because of the need to have both low bias and variance, it is difficult to optimize the model by estimating gradients between discrete steps. To simplify gradient optimization, rather than using discrete vectors, we introduce continuous and dense latent spaces. To obtain more diverse information, we introduce diversity learning based on the set of latent spaces $S\;=\;\{s_{n}^{k}\}\;$ (*k* = 1, 2, …, *K*; *K* being the number of latent spaces) and maximize their discrepancy during training. The spaces are assumed to be different multivariate Gaussian distributions about the image $I_{n}$ and the category $C_{n}$ with diagonal covariance. We use reparameterization to generate the means $M\;=\;\{\mu_{n}^{k}\}$ and standard deviations $\Sigma \;=\;\{\sigma_{n}^{k}\}$ of the spaces, which are calculated from $f_{n}^{J}$ using corresponding MLPs with fully connected layers of each space. Then, these means and standard deviations can be combined with the sampled unit Gaussian noise η as follows:(2)$$\begin{array}{cc} \; & s_{n}^{k}\;=\;\mu_{n}^{k}\;+\;\eta \sigma_{n}^{k}. \end{array}$$$f_{n}^{J}$ can be, respectively, mapped to the latent spaces as the latent matrix set $S\;=\;\{s_{n}^{k}\}\;$ to contain the information from the training data.

We use another MLP to obtain the reconstructed image features ${\hat{f}}_{n}^{I,k}$ from $s_{n}^{k}$ and then optimize the model by minimizing the following loss:(3)$$L_{I}\;=\sum_{k=1}^{K}\left|\right|f_{n}^{I}-{\hat{f}}_{n}^{I,k}{\left|\right|}_{2}.$$

Furthermore, we train the model by minimizing the Kullback–Leibler (KL) divergence, which is a regularizer in the original latent space that prevents the overfitting of the multi-latent space and ensures that the spaces generate a specific category of question. The KL divergence can be calculated using means and standard deviations as follows:(4)$$\begin{array}{cc} L_{\mathrm{KL}} & \;=\sum_{k=1}^{K}\mathrm{KL}\left(\mu_{n}^{k},\sigma_{n}^{k}\right)\\ & \;=\sum_{k=1}^{K}\frac{1}{2}\left(-\log{\left(\sigma_{n}^{k}\right)}^{2}\;+\;{\mu_{n}^{k}}^{2}+\;{\sigma_{n}^{k}}^{2}-\;1\right). \end{array}$$

Diversity learning ensures that the latent spaces contain diverse information. To make the model realize diversity learning, we translate diversity into the information difference between the latent spaces. When the information difference is significant, we assume that they contain diverse information. Therefore, we calculate the distribution similarity between the latent spaces and minimize their similarity during model training. To calculate the correlation of the distributions of the corresponding latent spaces, we introduce the MIC loss, which can be calculated as follows:(5)$$\begin{array}{cc} L_{\mathrm{MIC}} & \;=\sum_{k=1}^{K}\sum_{l=1}^{K}\mathrm{MIC}\left(s_{n}^{k},s_{n}^{l}\right)\\ & \;=\max_{s_{n}^{k}\times s_{n}^{l}<\lambda }\frac{I(s_{n}^{k},s_{n}^{l})}{\log_{2}\min\left(s_{n}^{k},s_{n}^{l}\right)},\\ I(s_{n}^{k},s_{n}^{l}) & \;=\;\int \int \;p\left(s_{n}^{k},s_{n}^{l}\right)\log_{2}\frac{p(s_{n}^{k},s_{n}^{l})}{p\left(s_{n}^{k}\right)p\left(s_{n}^{l}\right)}ds_{n}^{k}ds_{n}^{l}, \end{array}$$
where λ denotes a hyperparameter and $I(s_{n}^{k},s_{n}^{l})$ denotes an information coefficient. Because a lower $L_{\mathrm{MIC}}$ indicates that the latent spaces are more dissimilar, we train the informative multi-latent space by minimizing $L_{\mathrm{MIC}}$, and thus, different aspects of information are obtained.

The latent space set *S* is used in the following LSTM to generate the final question set. By using diversity learning, we contain diverse information from the training data in *S*, which is trained by $L_{\mathrm{KL}}$ and $L_{\mathrm{MIC}}$. Because the vectors of the multi-latent space contain abundant information, we can generate diverse questions from the latent spaces, which solves the informativeness problem.

### 3.3. Question Set Generation Based on LSTM

The outputs of previous models contain only a single question, and when generating multiple questions from a single image, these models are run repeatedly, resulting in many repetitions of the generated questions. To solve this problem, we simultaneously generate question sets with different meanings using the multi-latent space. The output question set $\langle q_{n}^{1},\;\ldots ,q_{n}^{K}\rangle$ is generated by LSTM based on $\left\{s_{n}^{1},\;\ldots ,s_{n}^{K}\right\}$ from the latent spaces, which can be expressed as
(6)$$\begin{array}{c} \langle q_{n}^{1},\;\ldots ,q_{n}^{K}\rangle \;=\;\langle \mathrm{LSTM}\left(s_{n}^{1}\right),\;\ldots ,\mathrm{LSTM}\left(s_{n}^{K}\right)\rangle . \end{array}$$

The generation loss of LSTM can be calculated as follows:(7)$$L_{G}\;=\sum_{k=1}^{K}\mathrm{CE}\left(q_{n}^{k},g_{n}^{k}\right),$$
where CE(·) denotes the cross-entropy loss function. We use the cross-entropy loss to control the quality of generated questions and to minimize the generation loss $L_{G}$ between the generated question $q_{n}$ and the ground truth $g_{n}$ during training.

### 3.4. Total Loss Function

The model’s final loss can be calculated as follows:(8)$$L\;=\;\alpha L_{I}\;+\;\beta L_{\mathrm{MIC}}\;+\;\gamma L_{\mathrm{KL}}\;+\;\delta L_{G},$$
where α, β, γ, and δ represent parameters that control the weights between the four loss functions. By training our model using the total loss function L, we can generate the category controllable question set.

## 4. Experiments

### 4.1. Conditions

Dataset. In our experiments, we used images and QA pairs from the VQA-RAD [40] and Slake [41] datasets simultaneously, and there are several QA pairs, as shown in Figure 2, for each image. VQA-RAD is a manually constructed medical-VQA dataset in radiology in which clinicians naturally create and validate questions and answers about images. Slake [41] selects healthy and unhealthy cases from open-source datasets [42,43,44]. Specifically, 179 chest X-Ray images are randomly selected with the original disease labels from [43], and 463 single-slice images from three-dimensional volume cases are chosen randomly from [42,44]. Then, experienced physicians label organs and diseases with ITK-SNAP [45] in as detailed a manner as possible.

In training, we chose six categories of QA pairs in the datasets, including “Abnormality,” “Modality,” “Organ,” “Plane,” “Position,” and “Others.” We computed the similarity between questions from one image using Google’s universal sentence-encoder [46] and selected questions with the same category and the lowest similarity as the ground truth. The details of our final dataset are presented in Table 1.

We randomly selected 100 images as the test set, and the remaining 856 images were divided into training and validation sets according to 5:1 during the training process; the training set contains 713 images, and the validation set contains 143 images.

We used the ImageNet pre-trained ResNet-50 [15] and LSTM as the image and language encoders, respectively, where LSTM had three layers and hidden layers with a size of 256. The details of the encoder and decoder are presented in Appendix A. In the experiment, the MLP we used had three layers and a hidden dimension of 512.

During training, we resized all images to 224 × 224 pixels and added a random flip. We set a learning rate of 0.001, which was reduced by one-tenth every five epochs until reaching $10^{-7}$, and the batch size was set to 32. We optimized the hyperparameters such that α=0.001, β=1, γ=0.001, and δ=1, and the weights of the loss functions were determined by selecting the best performance through experiments. All models were trained for 40 epochs, and the best results were used as the final results.

### 4.2. Evaluation Metrics

We used various metrics, including traditional evaluation metrics and our newly proposed metric, *Similarity*, to evaluate our model. The traditional metrics consisted of common language modeling evaluation metrics BLEU-*n* (*n* is 1 to 4) [26], METEOR [27], ROUGE [47], CIDEr [48,49], and *Inventiveness*. Language modeling metrics evaluate the accuracy of generated questions by comparing the directness of their words with that of the ground truth. Since a higher score means that the generated questions are closer to the ground truth, these quantitative evaluation metrics can directly reflect the accuracy of the generated sentences. *Inventiveness* indicates the percentage of generated questions completely different from the ground truth questions in the training set and is used to evaluate the diversity of VQG models. This evaluation metric reflects the diversity of the generated questions and proves that the evaluated model learns the knowledge in the dataset.

As depicted in Figure 3, because language modeling metrics evaluate generated sentences at the word level, the results will receive a higher score if they have more words in common with the ground truth, regardless of whether the meanings are similar or not, making it difficult to accurately evaluate the quality of the results. For example, *“Is there a liver in this image?”* and *“Does the picture contain liver?”* have the same meaning, but they obtain low scores (close to zero) for the language modeling metrics. On the other hand, *“Does the picture contain lung?”* and *“Does the picture contain liver?”* have entirely distinct meanings but have high scores greater than 0.8 for the language modeling metrics.

To avoid such a case, where inaccurate generation results have a high score, we introduce *Similarity* as a new metric. *Similarity* is the inner product of the semantic similarity of the features of sentences extracted by Google’s universal sentence-encoder-4 [46], which can determine whether sufficient information is contained in them. The first question pair above has a *Similarity* value of 0.77, showing that the two questions have a similar meaning, and the second pair’s *Similarity* value is 0.6, which is significantly lower than other language modeling metrics, indicating the information difference between the two questions. Existing evaluation metrics will fail in some special cases. Conversely, *Similarity* remains accurate.

In the experiments, we calculated the average *Similarity* between generated questions from each image as the *Similarity* between them and averaged the *Similarity* of all images as the final result. Low *Similarity* indicates that the generated questions contain more information.

### 4.3. Evaluation Results

To verify our model’s effectiveness, we compared our model with other state-of-the-art VQG models based on the VAE model. We first compared our model with the most original and simplest VQG model that uses a CNN and an RNN as a visual encoder and a language decoder, respectively. We also evaluated the performance of recently proposed models using the same VAE structure as ours, including medical VQG models: VQGR [21] and two general VQG models, C3VQG [36] and Krishna et al. [25]. These models use the VAE structure with the same architecture of encoder and decoder, and we also use the same format for the inputs and outputs. Thus, the improvement of the results can best prove the effect of the proposed multi-latent space and MIC loss function.

CNN + RNNmodel. 

The most basic generation model is based on the concept of VAE, using a CNN to extract image features and an RNN to generate sentences.

VisualQuestionGenerationfromRadiologyImages(VQGR). 

An approach to generate visual questions about radiology images, this method uses the VAE model to encode images into a latent space and decode natural language questions. In this method, the author also proposed a data augment method; however, to ensure the consistency of all models, we did not include the data augment part in the experiment.

InformationMaximizingVisualQuestionGeneration(Krishnaetal.). 

Krishna et al. [25] proposed a model based on VAE that attempts to learn a joint continuous latent space between the image, question, and the expected answer and maximizes mutual information from them. The method introduces a variational continuous latent space onto which the expected answers project. The proposed method can also maximize the evidence lower bound and control what information the generated questions request by reconstructing the image and expected answer representations.

CategoryConsistentCyclicVisualQuestionGeneration(C3VQG). 

A method based on Krishna et al. [25] that uses categories to constrain the generated questions, C3VQG proposed a novel category of consistent cyclic loss to enable the model to generate consistent predictions with respect to the answer category. Moreover, the proposed method imposes supplementary constraints on the latent space to provide a structure.

Table 2 shows the results of language modeling and diversity evaluation metrics. We applied our model using two, three, and four latent spaces, which are represented as **Our model-2**, **Our model-3**, and **Our model-4** cases, respectively. Our multi-latent space-based model outperformed previous models with only one latent space in most metrics. Specifically, we evaluated the diversity of our model using the new evaluation metric *Similarity*, and the results show that our model outperformed the previous methods. *Similarity* between the questions generated by our model was significantly lower than that of the other models, showing that the different latent spaces were considered to learn different knowledge during training and that the questions generated by our model contained more abundant information for a single image.

The results show that increasing the number of latent spaces within a certain range effectively improves the model’s performance, which proves the validity of our idea. However, due to the fixed amount of training data and the model’s complexity, blindly increasing the number of latent spaces is not feasible, and thus, the method of increasing the number of latent spaces has limitations. We believe that the most effective number of latent spaces will exist for a given model with a fixed amount of training data. According to the results in Table 2, we observe that accuracy begins to decrease when the number of latent spaces is four, and the highest accuracy is obtained when there are three latent spaces. **Our model-4** has the second-best results because the questions corresponding to the medical image contain various information aspects and require more than two latent spaces.

Here, we trained the model using four different loss functions with completely different weights. The weight was set by the performance of the verification set in the experiment. To verify the effectiveness of a single loss, we conducted ablation experiments. Except for the newly proposed MIC loss, the loss function had been extensively used in deep learning in the past, and many researchers, including VQGR [21], verified its effectiveness; thus, we only verified the newly proposed loss function in our experiments. Table 3 compares the model’s accuracy with and without the MIC loss. The results show that the model performed better with our proposed loss than without it, proving the effectiveness of the proposed loss function.

The results of tSNE [50] visualization of the embedding of the latent space vector set *S* are depicted in Figure 4. After the training process, our latent space vectors are mapped to different regions, which represent different knowledge obtained from the dataset. According to the visualization results of the embedding, the proposed methods can effectively learn different knowledge from different latent spaces and ensure the differences between them. The spatial differences demonstrated by the results also prove that our proposed MIC loss can effectively maximize the differences between different spaces. However, when there are four latent spaces, the gap between different latent spaces cannot be effectively controlled. We believe that this is because the questions in the datasets do not contain very different and diverse information, resulting in invalidation when the number of latent spaces is too large. Moreover, these results prove the limitations of the above-mentioned method. Since the number of latent spaces is adjustable, the results also indicate that there is an optimal number for different datasets. Our model can obtain good performance on different datasets by modifying the number of latent spaces.

We also evaluated the performance of our model qualitatively. In addition, to evaluate the *Similarity* between generated questions, we constructed a *Similarity* heatmap of the generated questions and the corresponding ground truth in Figure 5 as a sample. The generated questions with high *Similarity* with the ground truth demonstrate that our model has well contained the information in the training data. Furthermore, some questions have low similarity with the ground truth, showing that the generated questions contain many aspects of information about the target image. These informative questions will help solve the medical resource problems by reducing the involvement of human physicians in auxiliary clinical diagnosis. The questions similar to ground truth prove that our questions are meaningful. Compared with previous models that simply generate questions, the proposed method can generate questions with low similarity, allowing for more precise diagnoses by containing much information.

As another qualitative evaluation, Figure 6 also shows that our model can generate diverse question sets with low similarity under different latent spaces. During training, different latent spaces are considered to learn different knowledge, and by using these spaces, diverse questions can be generated during the generation process. From the generated results in Figure 6, for a single image, our model can simultaneously generate diverse questions with lower *Similarity* than the previous models. The diversity of the generated results can be reflected in two ways: the generation of different category questions for one image and the generation of questions with low similarity and containing various information for one category simultaneously. The experimental results quantitatively and qualitatively prove the diversity of the generated questions using our model. Furthermore, it shows that the informativeness problem can be solved.

## 5. Discussion

The proposed medical VQG model enhances the diversity of generated questions. In this section, we will discuss the limitations of the current model and future work.

### 5.1. Limitations

Although our results are more diverse and accurate than those of the previous model, there are still problems. The first is that the accuracy needs to be higher than that of other generation tasks and question generation using real-scene datasets. There are still many duplicates in the generated questions. The reason for this is that medical image processing requires a higher level of image understanding by the model, and the vocabulary in the medical domain is specialized. However, the number of medical datasets is significantly smaller than the real-scene datasets, making model training difficult. Furthermore, it is more challenging to control the categories of the medical questions than the real-scene data. This is because the medical questions are difficult to classify simply as the real scene, and the classification in the current dataset is not precisely parallel. There are many crossover cases, and a question could belong to more than one class simultaneously. This also makes model training difficult and leads to unsatisfactory results.

### 5.2. Future Work

In future work, we will improve the diversity and accuracy of the generated questions. In addition to expanding the dataset, there are two directions for improvement at the technical level: fine-tuning the pre-trained large-scale models and introducing new modalities. First, large-scale transformer-based image and language models [17,51] have performed better than traditional CNNs and RNNs on various tasks this year. Valanarasu et al. [52] and Gu et al. [53] have proposed transformer-based large-scale models for image and language modeling in the medical domains, respectively, with good performance, demonstrating the effectiveness of this concept. Thus, we will improve the performance of the models by using large-scale models. As human physicians typically refer to information outside the image when making a diagnosis, such as prior knowledge, Marino et al. [54] and Zheng et al. [55] have introduced outside knowledge to the VQA model, which also applies to VQG. By introducing outside knowledge, the accuracy of generated questions can be improved, and personalized question generation can be realized by introducing different knowledge, and this has good research prospects. Researchers have enhanced the visual reasoning ability by introducing semantic features [56], and improving the understanding of images can likewise generate more accurate questions. The question-and-answer-based interactive system can also be deployed on robots [57,58] to respond to complex and changing environments by increasing the flexibility of decision-making.

## 6. Conclusions

We have proposed a novel diversity learning-based VQG model with a multi-latent space for generating informative question sets from medical images. We introduce the MIC loss to train the multi-latent space and improve the question-generation process by simultaneously generating question sets. Furthermore, we introduce a new evaluation metric, namely, *Similarity*, to overcome the limitation of existing evaluation metrics. Although there are some problems with generating duplicate questions, our results are more accurate and diverse than those of previous models that used only a single latent space. We experimented with different latent spaces and proved that there was an optimal number of latent spaces for each dataset. Because the number of latent spaces varies, it can be adjusted to improve adaptability to different datasets. However, the proposed method has some limitations, and more studies are required to enhance diversity.

## Figures and Tables

**Figure 1 sensors-23-01057-f001:**
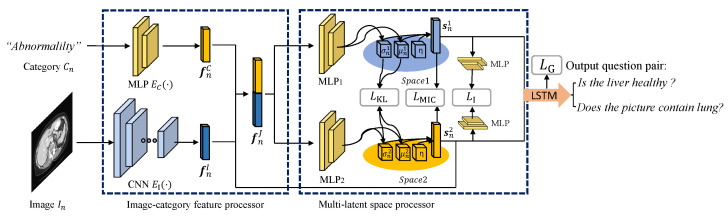
An overview of our medical VQG model with two latent spaces (1 and 2). The proposed model has two novel processors: an image-category feature processor and a multi-latent space processor. The input consists of a medical image and the category of the output question set. The image-category feature processor obtains and combines the features of the image and category. The multi-latent space processor ensures that different spaces contain different information. To make the overview more intuitive, we demonstrate the 2-latent space case.

**Figure 2 sensors-23-01057-f002:**
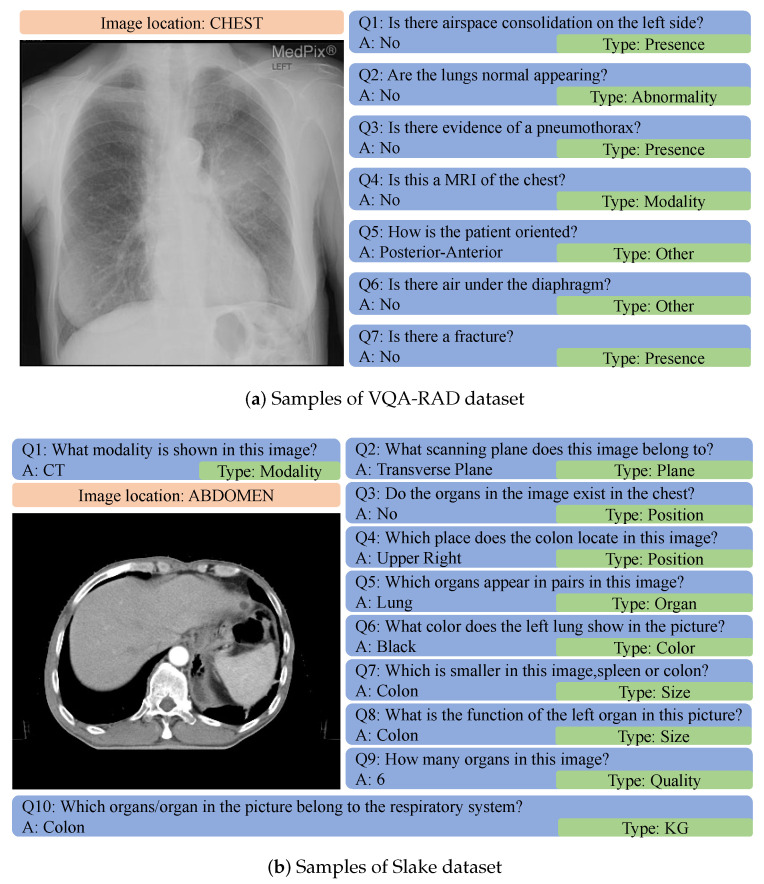
Samples of the datasets used in our experiments. Each image has several different types of QA pairs.

**Figure 3 sensors-23-01057-f003:**
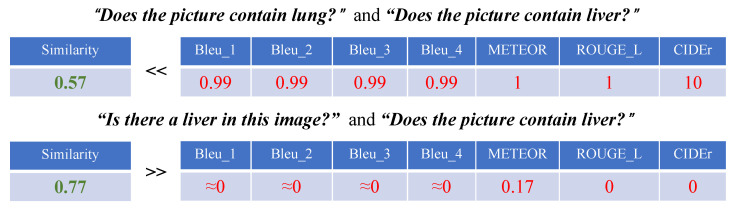
Example of the failure of traditional evaluation metrics. Although the previous indicators can evaluate the results to some extent, there are still many failures.

**Figure 4 sensors-23-01057-f004:**
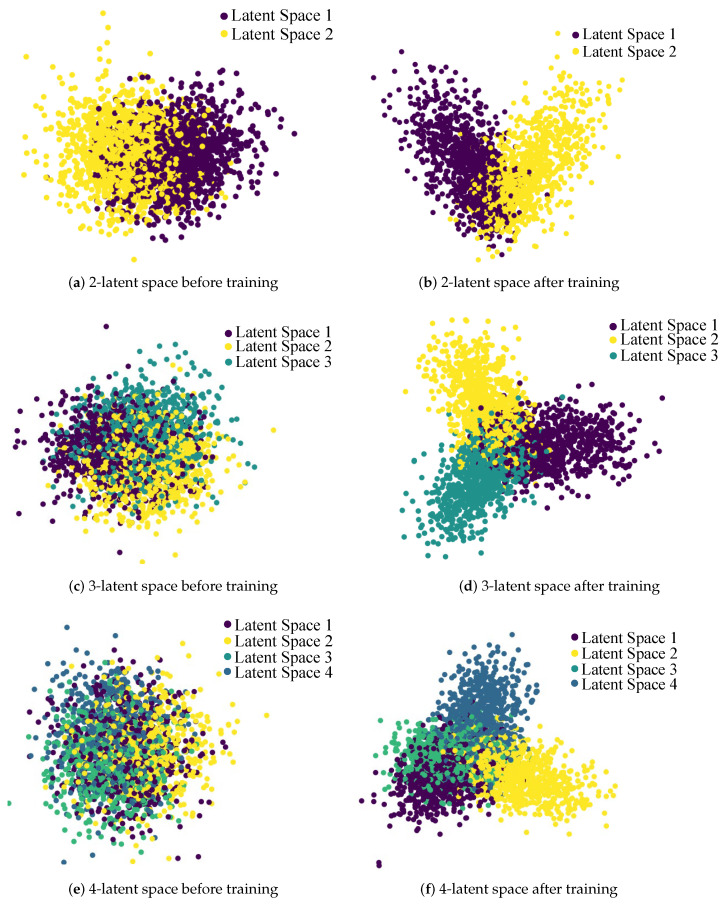
tSNE [50] visualization of the latent encodings of *S*. Different colors represent generated questions in different latent spaces, and we show the 2-, 3- and 4-latent space cases before and after training. We effectively separate the latent spaces through training. The encodings in different spaces are effectively separated in the 2- and 3-latent space cases.

**Figure 5 sensors-23-01057-f005:**
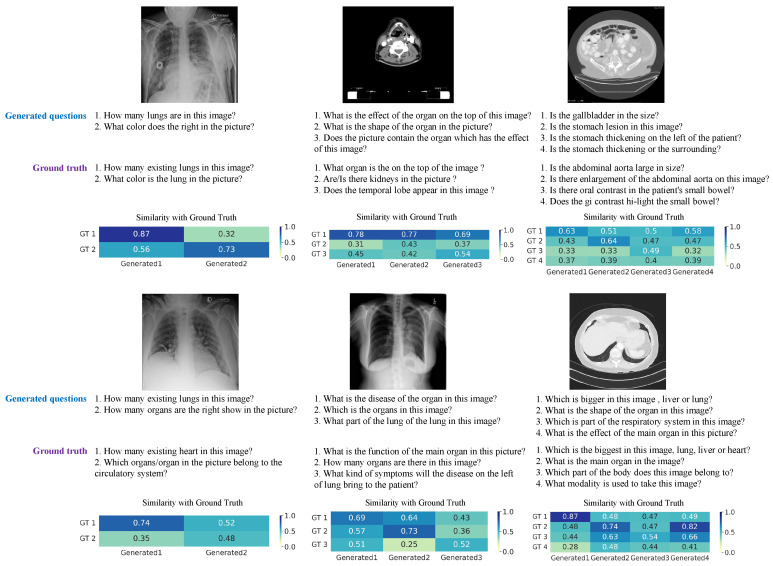
*Similarity* heatmap between the generated questions and the ground truth question sets.

**Figure 6 sensors-23-01057-f006:**
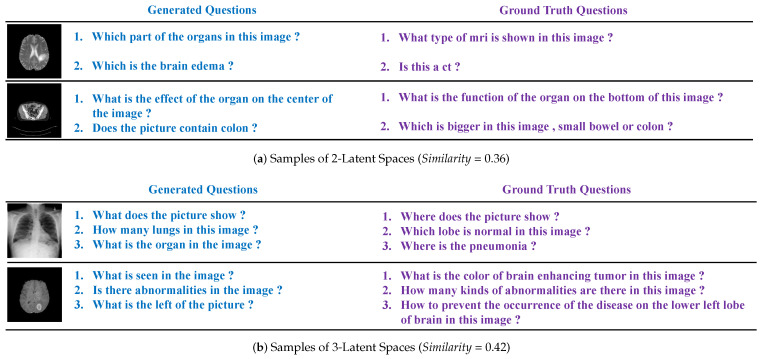
(**a**–**c**) are samples of the generated questions from a single image of our model with a different number of latent spaces. Questions generated from different spaces have lower *Similarity* than that of questions generated by the previous models shown in Table 2. Different spaces can be assumed to learn different knowledge, allowing for various questions to be generated.

**Table 1 sensors-23-01057-t001:** Details of the dataset we use after preprocessing.

Category	Question Number
Abnormality	312
Modality	290
Organ	290
Plane	484
Position	742
Others	1772

**Table 2 sensors-23-01057-t002:** Comparison of our model with other previous state-of-the-art models in terms of language modeling and diversity. These language modeling metrics represent the accuracy of the word level, and a high score indicates the model’s good performance. For the diversity metrics, high *Inventiveness* and low *Similarity* represent better performance. Our model variants, i.e., model-2, -3, and -4, represent models with 2, 3, and 4 latent spaces, respectively. The bolded numbers in the table are the best results, and the underlined numbers are the second best results.

	Language Modeling							Diversity	
	BLEU-1	BLEU-2	BLEU-3	BLEU-4	METEOR	ROUGE-L	CIDEr	*Inventiveness* (↑)	*Similarity* (↓)
CNN+RNN	38.3	22.2	14.6	10.4	15.4	37.8	55.4	0.10	0.61
VQGR [21]	40.3	23.8	15.7	11.3	15.9	39.7	64.1	31.1	0.65
C3VQG [36]	24.2	9.4	3.8	2.2	8.0	30.7	14.6	20.1	0.41
Krishna [25]	42.1	25.9	18.3	13.7	17.7	41.2	**83.1**	17.0	0.48
Our model-2	46.2	29.9	20.0	13.8	18.1	45.6	61.8	32.5	0.37
**Our model-3**	**48.7**	**33.0**	**23.2**	16.0	**19.6**	**50.0**	78.6	33.0	0.43
Our model-4	46.5	31.6	**23.2**	**16.8**	18.3	46.7	73.0	**33.9**	**0.35**

**Table 3 sensors-23-01057-t003:** Verification of the effectiveness of the newly proposed loss. “w M” denotes the model with the MIC loss, and the “w/o M” denotes the model without the MIC loss. The results show that the model’s accuracy is significantly improved with the addition of the new loss. The bolded numbers in the table are the best results.

	BLEU-1	BLEU-2	BLEU-3	BLEU-4	METEOR	ROUGE-L	CIDEr
Our model-2 w M	**46.2**	**29.9**	**20.0**	**13.8**	**18.1**	**45.6**	**61.8**
Our model-2 w/o M	42.6	26.4	16.3	11.2	16.5	43.1	44.2

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
