# Peer review of "Diversity Learning Based on Multi-Latent Space for Medical Image Visual Question Generation"

_sensors, 2023, doi:10.3390/s23031057_

Round 1
Reviewer 1 Report
The authors present a question generation system from medical images. The system is composed of an image feature extraction network to extract image features and a multi-layer perceptron to encode the category of the question. Features are used to discover a plurality of latent spaces, from whom a LSTM model can generate the question associated with the image and category. The key contribution of the paper is the design of a multi-term weighted cost function that takes into account the capacity of the latent space to represent the data, a regularization term to prevent over-fitting, a term to encourage diversity between latent spaces and a term measuring the similarity of questions generated from the latent space to that of the reference.
The paper is well written, and the experimental section is convincing.
The main criticism for the work comes from its conception. It is hard for this reviewer to understand its potential utility or applicability in any scenario of this work. What is the usefulness of generating the questions described in Figure 4?
The authors claim in lines 76-78 that VQA and VQG systems “provide systems with a clearer understanding of their illness to ease the tension among patients and possible misunderstanding about physicians”. Such affirmation may be true for VQA, however, it is very unclear how a VQG system, as the one proposed will fit such purpose.
Some minor typos in the paper:
Line 42. The acronym VOG is first used without description.
Line 82. Acronyms VOA and VOG used without description.
Line 83. Unclear what the authors refer to as “microscopic disease”
3.3 “question should be capitalized.
Reviewer 2 Report
This study have proposed a novel visual question generation model based on diversity learning with a multi-latent space for generating informative question sets from medical images. The research is significant and valuable. However, the content and organization of this paper can be improved.
1. The abstract should generally include the research background and purpose, research methods, research results, research importance and potential impact. The number of words should be controlled at about 250.
2. Each keyword is separated by a comma and does not need an "and" to connect.
3. How is the paper structured? Please add a paragraph to introduce it at the end of the introduction.
4. When the authors cite references, most of them are simply displayed in the paper. I suggest that the authors relate these references to the work of this paper, for example, how certain studies have influenced their work.
The following items are the references that been suggested:
https://doi.org/10.7717/peerj-cs.353
https://doi.org/10.1016/j.patcog.2021.108153
https://doi.org/10.3390/machines10010042
https://doi.org/10.1109/ACCESS.2021.3074937
https://doi.org/10.3390/s22062387
https://doi.org/10.3390/s22249733
5. How to ensure the fairness and effectiveness of quantitative evaluation?
6. How to ensure the richness of the final question set?
7. My suggestion is to move figure 6 into Appendix.
8. What is the limitations of this method?
9. What are the further research topics and directions? What else can be improved?
10. The conclusion of this paper needs to be optimized. We suggest that the author add some comparisons with previous work, advantages and disadvantages of the author’s method.
11. Please strictly typesetting the paper, especially the typography of the figures and tables.
Round 2
Reviewer 2 Report
The author's manuscript has undergone serious revisions, the quality has improved considerably, and it is recommended that it be considered for publication.